# A novel multi-agent simulation based particle swarm optimization algorithm

**Shuhan Du** *, **Wenhui Fan, Yi Liu**

Department of Automation, Tsinghua University, Beijing, China

* dush21@mails.tsinghua.edu.cn

## Abstract

Recently, there has been considerable research on combining multi-agent simulation and particle swarm optimization in practice. However, most existing studies are limited to specific engineering fields or problems without summarizing a general and universal combination framework. Moreover, particle swarm optimization can be less effective in complex problems due to its weakness in balancing exploration and exploitation. Yet, it is not common to combine multi-agent simulation with improved versions of the algorithm. Therefore, this paper proposes an improved particle swarm optimization algorithm, introducing a multi-level structure and a competition mechanism to enhance exploration while balancing exploitation. The performance of the algorithm is tested by a set of comparison experiments. The results have verified its capability of converging to high-quality solutions at a fast rate while holding the swarm diversity. Further, a problem-independent simulation-optimization approach is proposed, which integrates the improved algorithm into multi-agent systems, aiming to simulate realistic scenarios dynamically and solve related optimization problems simultaneously. The approach is implemented in a response planning system to find optimal arrangements for response operations after the Sanchi oil spill accident. Results of the case study suggest that compared with the commonly-used shortest distance selection method, the proposed approach significantly shortens the overall response time, improves response efficiency, and mitigates environmental pollution.

## Introduction

The merging of optimization and simulation technologies has seen rapid growth recently. The optimization of simulation models refers to finding the best values of some decision variables for a system where the performance is evaluated based on the output of a simulation model of this system [1]. However, due to the complexity, analytical simulation methods may be unsuitable for large-scale systems in manufacturing, supply chain management, financial management, etc. Agent-based modeling (ABM) is a bottom-up modeling method with high autonomy and interactivity, which is particularly suitable for complex system research [2]. Given the advantages of ABM, researchers have been solving optimization problems by integrating multi-agent simulation (MAS) with a wide range of methods, including game theory, reinforcement learning, and swarm intelligence (SI). Applications can be found in various

**Competing interests:** The authors have declared that no competing interests exist.

fields such as intelligent transport systems, efficient electric grids, and smart buildings. Among the methods mentioned above, SI has been proved to be robust and efficient for optimization problems. As a classic category of SI, particle swarm optimization (PSO) has the advantage of simple parameter setting, fast convergence rate, high stability, and strong scalability, making it an excellent option for complex systems.

However, despite a high convergence rate, the standard formation of PSO (SPSO) [3] is likely to stagnate at a local optimum, which reduces the solution accuracy. Thus, it is necessary to modify the algorithm for the application in complex problems where stable optimal solutions are indispensable. Different approaches have been considered to improve SPSO, among which setting parameters is especially representative [4]. Changing coefficients is a typical way of parameter setting. The performance of SPSO is affected by the values of its coefficients, i.e., the acceleration coefficients and the inertia weight. Thus, several attempts were made to tune the values of these parameters, such as the random weight method (RPSO) [5], the constriction factor approach (CPSO) [6], and the fuzzy adaptive approach (FAIPSO) [7]. Apart from the coefficients, the population size of the swarm is also a vital parameter. The basic idea for adjusting population size is to concentrate the individuals in the most promising area during the exploitation phase of the algorithm. De Oca et al. [8] conducted this idea by introducing incremental social learning (IPSO). On the other hand, as SPSO came from a simplified simulation of bird clustering behaviors, some researchers tried to improve it by mimicking other bird activities simultaneously. Neshat et al. [9] modified SPSO based on the predation phenomenon in bird swarms (PPSO). Besides these approaches, Li et al. [10] decoupled exploration and exploitation to make the algorithm more effective in large-scale optimization.

In MAS, agents can be either homogeneous or heterogeneous. Homogeneous agents are usually treated as particles when PSO is applied [11]. Yang et al. [12] presented a multi-agent-based model which simulated human behaviors in a multi-exit evacuation environment, where individuals were homogeneous agents and PSO was introduced to simulate their movement. Ali et al. [13] proposed a collective motion and self-organization control of a swarm of 10 unmanned aerial vehicles, which were divided into two clusters of five agents each. PSO was adopted to provide the best agents of the cluster. While homogeneous systems often simulate certain collective activities of agents with the same properties, heterogeneous systems are more practical for realistic scenarios due to the diversity of agents. Kanaga and Valarmathi [14] proposed a multi-agent model using PSO to solve patient scheduling problems, using a PSO Agent to perform the algorithm. Yu et al. [15] designed a Web services selection model, using PSO to select the optimal Web services combination in E-business. The model was realized by MAS. Thiel et al. [16] simulated Hanoi dwellers' choices between traditional markets and a hypothetical new one, aiming to find the optimal location for the ready-to-build local supermarket by using PSO to maximize sales volume. Mahad et al. [17] proposed an intelligent MAS approach to optimize power supply in community-based multi-microgrids systems, where various layers of autonomous and intelligent agents took decisions based on the PSO method.

Although there are many studies on improving SPSO, there are not so many attempts to combine proposed algorithms with MAS and apply them to practical problems. In fact, utilizing improved algorithms in simulation-optimization systems is necessary because complex tasks require high-performance optimizers, and SPSO may not meet this requirement. Besides, existing studies on the combination of MAS and PSO may vary in engineering fields, yet they all designed the system based on specific problems of their own research field without summarizing a general and universal framework, which is field-independent and can be implemented in various settings. Therefore, this paper focuses on a more in-depth exploration of improving

PSO and integrating it into MAS to solve practical problems. The main contributions of this paper are as follows:

1. Proposing an improved particle swarm optimization algorithm (CoPSO) according to the biological nature of birds. It has a multi-level structure and a competition mechanism, aiming to improve performance by enhancing exploration while balancing exploitation.

2. Proposing a general simulation-optimization approach (MAS-CoPSO) to integrate CoPSO into MAS. It is supposed to achieve a very close combination between simulation and optimization, and the application should not be limited to certain engineering fields and specific problems.

3. Conducting a series of experiments to substantiate the validity of CoPSO.

4. Implementing MAS-CoPSO in a case study of the Sanchi oil spill accident to solve the response planning problem.

## Methods and materials

This section elaborates on the methodologies used in this study. Firstly, a brief introduction to MAS technology is given. Detailed descriptions of the proposed CoPSO algorithm and the MAS-CoPSO approach follow afterward.

### Multi-agent simulation

The fundamental element in MAS is the agents, which can be software or physical entities. Although different agents exhibit distinct behaviors, they share some common properties. For instance, they all have a certain degree of autonomy that enables them to work even without human intervention, they can communicate and interact with each other, and they are capable of perceiving and reacting to the changes in the environment as well as determining the proper behaviors to achieve the final goal [18]. In a simulation system, agents can be either homogeneous or heterogeneous depending on the specific problem background. Heterogeneous agents can deal with diverse scenarios and complex tasks, making it more suitable for complicated realistic systems [11]. Based on the aim of proposing a general framework for the combination of MAS and CoPSO, this paper mainly focuses on heterogeneous systems due to their universality.

### Improved particle swarm optimization algorithm

PSO mimics the behavior of bird swarms. The fundamental idea is to assume a potential solution to the optimization problem as a bird without quality and volume, i.e., a particle. Particles are described by their positions and velocities. Generally, the position of a particle represents a specific solution while the velocity represents the searching direction and scope, which leads the particle to fly through the solution space. Every particle will get a fitness value by calculating the objective function. It will also adjust the value according to the experience of itself and the neighbors.

Supposing the search space is D-dimensional, the $i$th particle is represented as $\mathbf{x_i} = (x_i^1, x_i^2, \ldots, x_i^D)$, where $x_i^d \in [X_{min}^d, X_{max}^d]$, $d \in [1, D]$, $X_{min}^d$ and $X_{max}^d$ are the lower and upper bounds of the $d$th dimension, respectively. The velocity of the $i$th particle is represented as $\mathbf{v_i} = (v_i^1, v_i^2, \ldots, v_i^D)$, where $v_i^d \in [V_{min}, V_{max}]$, $d \in [1, D]$, $V_{min}$ and $V_{max}$ are the minimum velocity and maximum velocity specified by the user, respectively. In SPSO [3], during each

iteration, particle $i$ updates its position and velocity as follows:

$$\mathbf{v_i} \leftarrow w\mathbf{v_i} + r_1 c_1 (\mathbf{pbest_i} - \mathbf{x_i}) + r_2 c_2 (\mathbf{gbest} - \mathbf{x_i}), \tag{1}$$

$$\mathbf{x_i} \leftarrow \mathbf{x_i} + \mathbf{v_i}, \tag{2}$$

where $w$ is the inertia weight, $r_1$ and $r_2$ are random values between 0 and 1, $c_1$ and $c_2$ are acceleration constants which control how the particle moves, $\mathbf{pbest_i} = (pbest_i^1, pbest_i^2, \ldots, pbest_i^D)$ is the best previous position of the $i$th particle, and $\mathbf{gbest} = (gbest^1, gbest^2, \ldots, gbest^D)$ is the best position among all the particles in the swarm.

PSO encourages exploration during the early stage of the iterations while encouraging exploitation in the latter stage. However, Angeline [19] has pointed out that the balance between exploration and exploitation is subtle and difficult to control in PSO, leading to stagnation or premature convergence. Inspired by the features of bird flocks, this paper proposes CoPSO to improve the performance of PSO, introducing a multi-level structure and a competition mechanism.

**Multi-level structure.** According to the biological nature of birds, their ability to sing and create a complete song is learned from their parents. Experiments have shown that if a bird is reared in silence, it can only scream. In addition, birds' singing skills are not set in stone. Basically, as they grow up, their expertise in singing also gets improved [20]. Based on this phenomenon, CoPSO has a multi-level structure, dividing the particles in a swarm into different singing levels. It is stipulated that the closer a particle is to a possible global optimum, the higher its level. The level of the $i$th particle is denoted by $l_i$, which will increase if the particle doesn't update its personal best position $\mathbf{pbest_i}$ in $\mu$ continuous iterations. Since birds of different ages learn to sing at different rates, particles of different levels have different acceleration constants.

Fig 1 illustrates how the particles' levels change during the iterations in two-dimensional unimodal problems. At the early stage of the algorithm, particles are like newborn birds with little knowledge about singing. They will explore the search space and try to find promising areas. Once found, they might temporarily stop updating their personal best positions, leading to an upgrade. At the latter stage of the algorithm, some particles can be very close to the global optimum. Consequently, they will find it difficult to locate better positions, and their levels are also relatively higher.

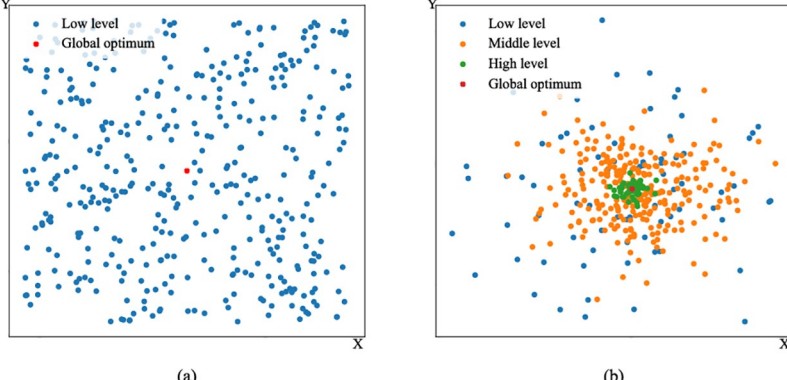

(a)                                                             (b)

**Fig 1. Variations of particles' levels during the iterations in two-dimensional unimodal problems.** The points represent the particles. (a) The early stage. (b) The latter stage.

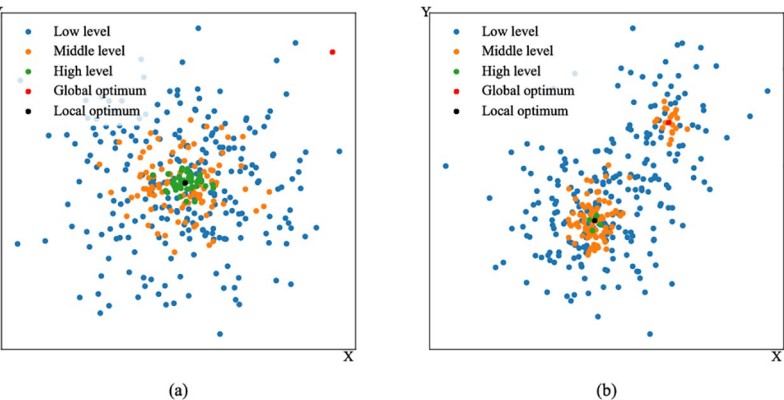

**Fig 2. Competition mechanism in two-dimensional multimodal problems.** The points represent the particles. (a) Particles are trapped in the local optimum. (b) Particles jump out of it.

**Competition mechanism.** While a particle's high level is likely to come from finding the global optimum, there are exceptions. Note that falling into local optima can also stop particles from updating their personal best positions. Since particles' levels are linked to their ages (the higher level and the more skillful in singing, the older), the idea of the competition mechanism inside bird swarms is utilized to improve the algorithm. The survival resources of swarms and the lifespan of birds are both limited. Therefore, a bird swarm is constantly renewed by the death of old birds and the born of new ones. If a particle's level is too high, it becomes too old to possess the resources of the swarm. The population size is set as a constant, once a particle's level reaches the upper bound $L_{max}$, it will be replaced by a randomly generated new one. The new particle has the lowest level since it's a newborn.

Fig 2 depicts how the competition mechanism functions in two-dimensional multimodal problems. If particles prematurely converge to a local optimum, they will upgrade around it. It can be seen from Fig 2a that few particle gets close to the global optimum. However, once the particles in the local area become too high-leveled, they will be replaced by new birds. As these new particles are randomly generated, they have the potential to explore unknown search space and finally locate the global optimum, which is shown in Fig 2b.

Since particles tend to move closer to the global optimum and oscillate around it as the algorithm runs, in practice, the value of $\mu$ will be increased as the level rises, which allows an elaborate control over the balance between exploration and exploitation. The pseudo-code of CoPSO is shown in Algorithm 1.

**Algorithm 1**: Pseudo-code of CoPSO

```
Input: The value of Lmax, the value of μ, and the maximum number of
iterations Imax
Output: The final solution gbest and its fitness value f(gbest)
1 Initialize gbest;
2 for each particle i do
3   Generate xi and vi randomly;
4   pbesti = xi;
5   if f(xi) is better than f(gbest) then
6     gbest = xi;
7   end
8 end
9 while Imax is not met do
10   for each particle i do
11     Update vi according to Eq (1);
```

```
12      Update xi according to Eq (2);
13      if f(xi) is better than f(pbesti) then
14        pbesti = xi;
15      end
16      if f(xi) is better than f(gbest) then
17        gbest = xi;
18      end
19      if pbesti hasn't changed in μ continuous iterations then
20        li+ = 1;
21      end
22      if li == Lmax then
23        Regenerate xi and vi randomly;
24        li = 0;
25      end
26    end
27 end
```

## Problem-independent simulation-optimization approach

The architecture of the proposed MAS-CoPSO approach is shown in Fig 3, where the MAS module and the CoPSO module are the main components. For the optimization problem brought up by the user (e.g., decision making, scheduling, route planning, etc.), a suitable simulation time step should be chosen first according to specific backgrounds. In every time step, MAS is supposed to simulate the problem scene, during which the agents' behaviors might change the values of related variables and parameters. Simulation statistics are then put into

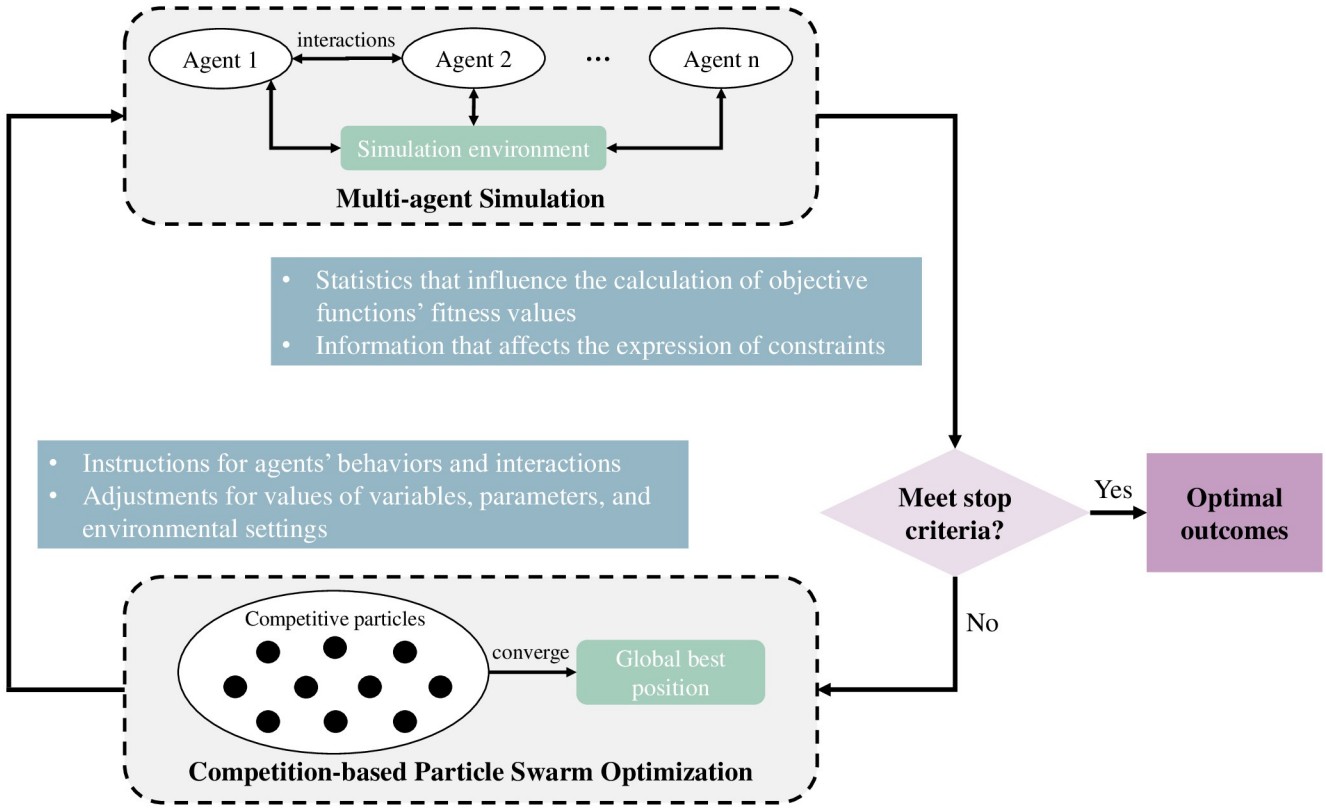

**Fig 3. Architecture of the MAS-CoPSO approach.**

the optimization module, whose objective can be maximizing the efficiency or minimizing the cost of certain activities. CoPSO algorithm is supposed to optimize the objective function while satisfying all the constraints. The optimization result will then play as the input of MAS, instructing the simulation process in the next time step. This iteration will go on until meeting the stop criteria set by the user. The outcome of MAS-CoPSO is the final solution to the problem. For complex systems, the solution may not be strictly optimal (actually it is intractable to find the strict global optimum for these problems). However, MAS-CoPSO can compromise between the solution quality and computational cost, making it possible to get a satisfying result in a reasonable period of time.

Successful simulation of the problem scene relies on a proper setup of the environment (which reflects the background of the problem) and an appropriate design of all the agents. In MAS, agents have autonomous behaviors and complex interactions, and the relationship between them can be various (e.g., cooperation, competition, control, etc.). Value changes of related variables and parameters link simulation and optimization together. For CoPSO, simulation results will influence the calculation of the objective function's fitness value and the expression of constraints. Optimization results of CoPSO will function as the instructions for simulation in the next time step, i.e., how to adjust agents' behaviors and tune the values of variables, parameters, and environmental settings. These two modules compose a cohesive framework through consistent information exchange and close integration.

## Experiments and analysis of the optimization algorithm

To substantiate the validity of the proposed CoPSO algorithm, this study conducts numerical comparison experiments and computational complexity analysis. Exhaustive descriptions and discussions are given in this section.

### Comparison experiments

The performance of CoPSO is compared with a set of classic algorithms:

- **SPSO** [3]: the standard formation of PSO, introduced in detail in the previous section.

- **RPSO** [5]: PSO with a random inertia weight factor designed for tracking dynamic systems.

- **CPSO** [6]: PSO with a new constriction factor related to the original acceleration coefficients.

- **IPSO** [8]: PSO with an increasing population size, i.e., whenever the algorithm can not find a satisfactory solution, add a new particle to the population.

**Benchmarks and parameter setting.**   Five well-known benchmark functions commonly used in literature [21] are applied to evaluate the performance of CoPSO, both in terms of solution quality and convergence rate. They are non-linear minimization problems that present different difficulties to the optimizers. Table 1 lists the functions, the problem dimension $n$, the global minimum fitness value $f_{min}$, and the search space ranges (which are also the initial ranges in this research).

The main parameters are set based on the recommendation in [22]. In CPSO, the maximal velocity $V_{max}$ and the minimal velocity $V_{min}$ are 100,000 and -100,000, respectively (since it is believed that $V_{max}$ and $V_{min}$ are not even needed in CPSO [6]). For other algorithms, $V_{max}$ is 10% of the search space's upper bound while $V_{min}$ is 10% of the lower one. The acceleration constants $c_1$ and $c_2$ are both 2.05 for CPSO. For SPSO, RPSO, and IPSO, the figure is 2.0. For CoPSO, $L_{max}$ is 3, the initial value of $\mu$ is 5 (added by 1 for each level up), and the gap between

**Table 1. Benchmark functions used in this study.**

| | Function | $n$ | $f_{min}$ | Search range |
|---|---|---|---|---|
| Sphere function | $f_1(x) = \sum_{i=1}^n x_i^2$ | 30 | 0 | $[-100, 100]^n$ |
| Schwefel's function | $f_2(x) = \sum_{i=1}^n |x_i| + \prod_{i=1}^n |x_i|$ | 30 | 0 | $[-10, 10]^n$ |
| Quartic function | $f_3 = \sum_{i=1}^n i x_i^4 + rand[0, 1)$ | 30 | 0 | $[-1.28, 1.28]^n$ |
| Rosenbrock function | $f_4 = \sum_{i=1}^n 100 \times (x_{i+1} - x_i^2)^2 + (1 - x_i)^2$ | 30 | 0 | $[-100, 100]^n$ |
| Griewank function | $f_5 = \dfrac{1}{4000} \sum_{i=1}^n x_i^2 - \prod_{i=1}^n \cos\left(\dfrac{x_i}{\sqrt{i}}\right) + 1$ | 30 | 0 | $[-600, 600]^n$ |

The problem dimension $n$, the global optimum $f_{min}$, and the search range are listed in the table.

the acceleration constants of each level is 0.2 ($c_1$ and $c_2$ are the same with a maximum value of 2.0). A fixed inertia weight value of 0.6 is adopted for SPSO, IPSO, and CoPSO. The maximum iteration number is 1000 for all algorithms, meaning that the maximum particle number is 1000 in IPSO. In other cases, the population size is 80. A total of 50 repeating runs are conducted for each experiment. In addition, as recommended by Yang et al. [23], Wilcoxon rank sum tests are conducted for the statistical analysis between CoPSO and the other algorithms with a significance level of 0.05.

**Results and discussion.** Table 2 summarizes all the experimental results of 50 independent runs. Wilcoxon rank sum tests further verify the significance of these numerical results. The numbers in bold represent the comparatively best values. In general, CoPSO greatly outperforms SPSO, RPSO, CPSO, and IPSO on most metrics. For the exceptions, it still shows equivalent performances to the comparison algorithms. These results indicate that the multi-level structure and the competition mechanism can improve the solution quality of CoPSO.

It is believed that CoPSO can preserve excellent exploration and exploitation abilities while keeping a good balance between these two. In other words, CoPSO is able to jump out of local optima and fully exploit the promising areas at a fast rate. To substantiate this hypothesis, swarm diversity is used to measure an algorithm's exploration ability. Supposing the size of swarm $S$ is $N$, the problem dimension is $D$, and $x_i$ represents the position of particle $i$. As recommended by Yang et al. [23], the diversity of swarm $S$ is computed as follows:

$$D(S) = \frac{1}{N} \sum_{i=1}^N \sqrt{\sum_{d=1}^D (x_i^d - \bar{x}^d)^2},$$

$$\bar{x}^d = \frac{1}{N} \sum_{i=1}^N x_i^d,$$

where $D(S)$ is the swarm diversity of $S$ and $\bar{x}$ is the average position of the swarm. Further, the converging speed can reflect whether an algorithm can compromise between exploration and exploitation properly. During the iterations, average swarm diversities and average global best fitness values (of 50 runs, in logarithmic form) for each algorithm on five benchmarks are recorded in Fig 4.

Generally, CoPSO converges faster with better solutions than all other algorithms in most cases. As for swarm diversity, CoPSO holds a similar tendency on all benchmarks, i.e., decreasing rapidly in the early stage of the evolution and staying at a relatively high value (or even the highest) in the latter stage. This can be explained by the structure of CoPSO. Particles of different levels are in different regions of the search space. Those of lower levels are more likely to

**Table 2. Experimental results for all algorithms on benchmark functions.**

|  |  | CoPSO | SPSO | RPSO | CPSO | IPSO |
|---|---|---|---|---|---|---|
| $f_1$ | Best | **2.8549E-65** | 5.5264E-09 | 1.5411E-35 | 1.5978E-23 | 3.0563E-11 |
|  | Worst | **9.7607E-57** | 5.7243E-06 | 6.5542E-29 | 1.2931E-19 | 7.2027E-09 |
|  | Median | **1.0691E-61** | 8.5793E-08 | 5.3866E-33 | 9.3553E-22 | 3.6676E-10 |
|  | Mean | **3.0003E-58** | 4.1779E-07 | 1.3600E-30 | 7.2911E-21 | 7.5476E-10 |
|  | Std | **1.4938E-57** | 9.5508E-07 | 9.2625E-30 | 2.1853E-20 | 1.2102E-09 |
|  | p-value | - | 7.0661E-18 | 7.0661E-18 | 7.0661E-18 | 7.0661E-18 |
| $f_2$ | Best | **4.9143E-37** | 3.2218E-05 | 1.2864E-21 | 5.8821E-11 | 3.5853E-06 |
|  | Worst | **8.5886E-15** | 9.8248E+00 | 6.4473E-13 | 6.8001E-07 | 2.1356E+02 |
|  | Median | **1.3265E-31** | 3.1720E-04 | 1.0877E-19 | 2.6652E-09 | 9.8882E-05 |
|  | Mean | **3.8317E-16** | 3.9115E-01 | 1.6602E-14 | 3.7949E-08 | 7.5323E+00 |
|  | Std | **1.4994E-15** | 1.6359E+00 | 9.2441E-14 | 1.1971E-07 | 3.3733E+01 |
|  | p-value | - | 7.0661E-18 | 5.1559E-11 | 7.0661E-18 | 7.0661E-18 |
| $f_3$ | Best | 3.0297E-03 | 2.3906E-02 | 8.9131E-03 | **2.7188E-03** | 7.4989E-03 |
|  | Worst | **1.7855E-02** | 1.3631E-01 | 5.5517E-02 | 2.1025E-02 | 3.4737E-02 |
|  | Median | 8.6013E-03 | 7.1187E-02 | **4.0110E-03** | 1.0032E-02 | 1.7729E-02 |
|  | Mean | **8.7629E-03** | 7.3491E-02 | 2.5168E-02 | 1.0271E-02 | 1.8165E-02 |
|  | Std | **3.4998E-03** | 24612E-02 | 9.1510E-03 | 4.6240E-03 | 6.8623E-03 |
|  | p-value | - | 7.0661E-18 | 3.7961E-13 | 1.0448E-01 | 3.9592E-12 |
| $f_4$ | Best | **3.4746E-03** | 1.4929E+01 | 6.4345E+00 | 7.6875E+00 | 4.0219E+00 |
|  | Worst | **8.1844E+01** | 4.6145E+02 | 1.5713E+02 | 9.5701E+02 | 1.7156E+02 |
|  | Median | **1.3684E+01** | 7.2457E+01 | 2.4572E+01 | 2.5592E+01 | 2.6071E+01 |
|  | Mean | **1.7385E+01** | 6.7829E+01 | 4.4824E+01 | 8.7070E+01 | 4.4996E+01 |
|  | Std | 1.8940E+01 | 6.8713E+01 | 3.3475E+01 | **1.5140E+01** | 3.6439E+01 |
|  | p-value | - | 3.8499E-14 | 6.9808E-10 | 3.7706E-12 | 3.5908E-12 |
| $f_5$ | Best | **0** | 3.3119E-10 | **0** | **0** | 9.4358E-13 |
|  | Worst | **2.9459E-02** | 5.1369E-02 | 4.4293E-02 | 6.6471E-02 | 5.3964E-02 |
|  | Median | 7.3960E-03 | 7.3961E-03 | **7.3960E-03** | **7.3960E-03** | 9.8573E-03 |
|  | Mean | **7.1907E-03** | 9.0566E-03 | 9.6510E-03 | 1.0782E-02 | 1.1233E-02 |
|  | Std | **8.2867E-03** | 1.1201E-02 | 1.1035E-02 | 1.3882E-02 | 1.0670E-02 |
|  | p-value | - | 1.0732E-02 | 3.8092E-01 | 3.7739E-01 | 8.9401E-04 |

Best, Worst, Median, Mean, and Std are the best, worst, median, average, and the standard deviation of the final results in 50 independent runs, p-value is the statistical result obtained by Wilcoxon rank sum test with a significance level of 0.05.

be far away from the global best position while those of higher levels can be very close to it. CoPSO treats different levels differently by enhancing exploration for lower levels and promoting exploitation for higher ones. Thus, the algorithm has more potential to locate promising areas in the early stage and then converges to them very quickly, causing the swarm diversity to decline. However, particles may not upgrade because they find a global optimum, but because they are trapped in local optima. In this case, the competition mechanism will force the swarm to escape from local areas, which increases the diversity simultaneously.

Despite the overall similarity, the behaviors of CoPSO differ slightly on different kinds of test functions. $f_1$ is a simple unimodal function with only one global minimum, making it possible to achieve fast convergence. Although all algorithms converge exponentially to the optima, CoPSO greatly surpasses the others due to better exploitation of the promising areas at the beginning and high-intensity exploration in the latter stage. For $f_2$, CoPSO has a

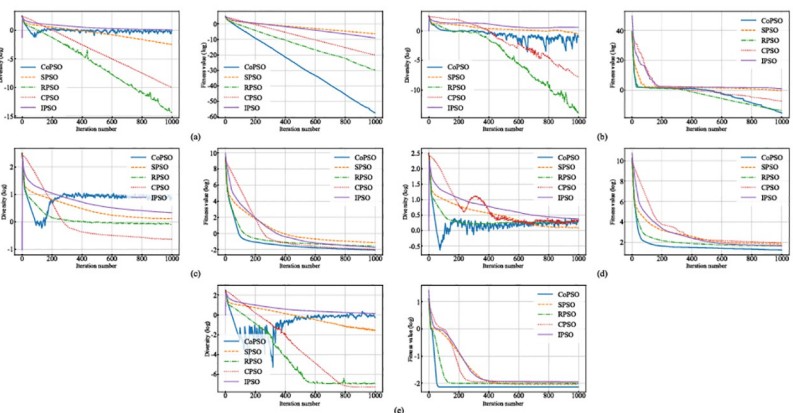

**Fig 4. Average swarm diversities and average global best fitness values of all algorithms at each iteration.** Note that the vertical axes are in logarithmic form. (a) The results on $f_1$. (b) The results on $f_2$. (c) The results on $f_3$. (d) The results on $f_4$. (e) The results on $f_5$.

competitive performance compared to RPSO and even shows better potential in the end as a result of higher swarm diversity. $f_3$ is a noisy quartic function. $f_4$ is a classic optimization problem with a global minimum inside a long, narrow, parabolic-shaped valley. $f_5$ is a multimodal function where the number of local optima increases exponentially as the problem dimension increases. For all these complicated benchmarks, CoPSO always keeps an excellent balance between exploration and exploitation, which helps it to achieve the fastest converging rate or the shortest stagnation at local optima. The other algorithms lack the adjusting ability of CoPSO. For instance, IPSO puts too much emphasis on exploration (holding the highest swarm diversity on most benchmarks) while RPSO and CPSO are too focused on exploitation, which degenerates their overall performances.

## Computational complexity analysis

According to Yang et al. [23], given a fixed number of fitness evaluations, the computational complexity of an evolutionary algorithm is generally calculated by analyzing the extra cost in each generation without considering the cost of function evaluations, which is problem-dependent. As CoPSO inherits the simple structure of SPSO, its computational complexity will be analyzed by comparison with SPSO.

Assuming the population size is $N$ and the problem dimension is $D$, updating particles' states takes $O(N \times D)$ time in each iteration for SPSO. By comparison, adding line 19 to line 25 in Algorithm 1 costs extra time for CoPSO. Specifically, particles tend to update **pbest$_i$** frequently at the early stage of the iterations, so it only costs $O(N)$ extra time to check $l_i$ and update related parameters. As $l_i$ increases, the algorithm may have to regenerate some new particles, which costs $O(N \times D)$ extra time in the worst case. However, this situation is only

**Table 3. Average runtime for all algorithms on benchmark functions (in second).**

|  | CoPSO | SPSO | RPSO | CPSO | IPSO |
|---|---|---|---|---|---|
| $f_1$ | 9.3840e-02 | 9.2220e-02 | 9.0920e-02 | 8.9540e-02 | 5.3714e-01 |
| $f_2$ | 9.6400e-02 | 9.2880e-02 | 9.3700e-02 | 9.0220e-02 | 5.6548e-01 |
| $f_3$ | 1.5344e-01 | 1.5248e-01 | 1.5138e-01 | 1.5044e-01 | 9.3878e-01 |
| $f_4$ | 1.0150e-01 | 9.8220e-02 | 9.9000e-02 | 9.8580e-02 | 5.7976e-01 |
| $f_5$ | 1.5624e-01 | 1.5144e-01 | 1.5274e-01 | 1.5212e-01 | 9.3274e-01 |

possible in certain generations. Besides, the value increase of $\mu$ also controls the time cost of this part. As for the space complexity, since CoPSO needs to store the information of $l_i$ and the time particles stay at the same **pbest$_i$**, it requires $O(N)$ extra memory than SPSO (which takes $O(N \times D)$ space to save particles' positions and velocities). To conclude, with proper parameter setting, the computational complexity of CoPSO can be controlled within an acceptable range.

Table 3 lists the average computing time for a single run (in 50 repeating experiments) with the same parameters as the numerical experiments. SPSO, RPSO, and CPSO take nearly the same time on different functions due to their similar algorithmic structures. However, IPSO has a growing population, dramatically increasing fitness evaluations at the latter stage of the iterations. Consequently, the computational cost of IPSO is the highest. Based on the complexity analysis above, CoPSO takes acceptable extra time in each iteration compared to SPSO, verified by the experimental result that CoPSO costs slightly higher than SPSO on the benchmarks. In conclusion, CoPSO surpasses the other algorithms in solution quality while remaining computationally efficacious.

## Case study of the simulation-optimization approach

To demonstrate that MAS-CoPSO can be used in practical applications, a case study of a real marine oil spill accident is carried out. This section presents the accident background, the implementation of MAS-CoPSO in a response planning system, and the mathematical formulation of the optimization problem. Further, an in-depth analysis of the system's performance is given.

### Background

Frequent marine oil spills have posed severe tests to the environment. Reducing the loss caused by oil spills has become an important issue. The capability of the MAS-CoPSO approach is tested through a case study of the Sanchi oil spill accident, which is caused by a collision between Panamanian oil tanker Sanchi and Chinese bulk carrier Changfeng Crystal. The accident happened on January 6, 2018, and the collision site is about 160 nautical miles east of the Yangtze River Estuary. Fig 5 illustrates the approximate location where the accident happened

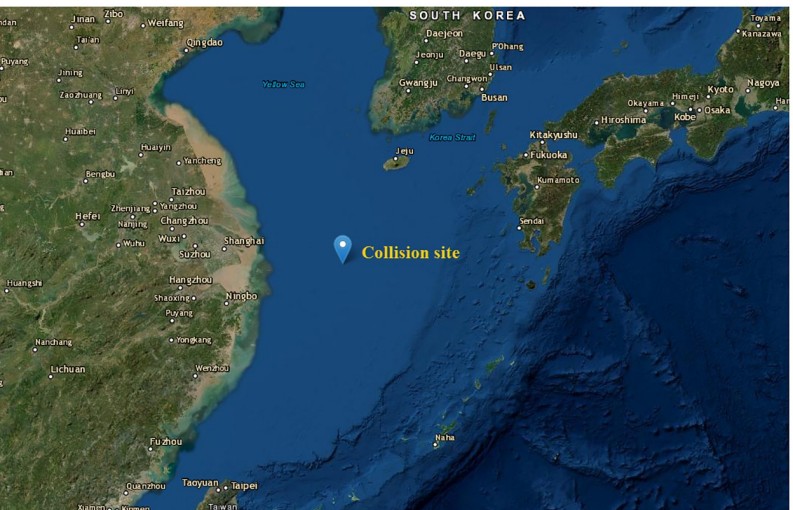

**Fig 5. Geographical map of the approximate collision site and surrounding areas.** Map services and data available from the U.S. Geological Survey.

and the areas around it. Tons of condensate oil carried by Sanchi leaked into the ocean, causing significant economic loss and environmental pollution. Researchers have pointed out that oil tankers are transporting around 90% of all the oil around the world [24]. Furthermore, ship collision is the primary cause of many catastrophic marine oil spills, which suggests the representativeness of the case study.

Zhong and You [25] have concluded that after spill accidents, oil leaked into the ocean forms scattered slicks around the spill site and undergoes various physical and chemical changes due to environmental forces. The most dominant processes are spreading, evaporation, emulsification, and dispersion. These processes will dramatically change the volume, area, thickness, and viscosity of those slicks, which will affect the efficiency of oil spill response operations as a consequence.

There are some equipped response teams berthed at docks near the spill site. They will conduct a wide range of actions to clean up the ocean, among which booms, skimmers, chemical dispersants, and in situ burning are the most frequently used methods. Booms are generally the first equipment deployed after an oil spill and are often used to protect shorelines, divert oil to certain areas, or concentrate oil for further recovery and burning. Skimmers are adopted to recover oil or oil-water mixtures from the water surface. The characters of the slick will affect the working efficiency of skimmers. Chemical dispersants can reduce the oil-water interfacial tension, which should be initialized as soon as possible. In situ burning indicates controlled burning of the oil at or near the site. In this case, it is assumed that the spraying of dispersants and the concentration of leaked oil have been completed quickly after the accident. Besides, in situ burning will emit toxic substances and waste the leaked oil. Thus, the primary response operation is recovery.

This study focuses on supporting response planning after the accident. A simulation-optimization system is built based on the proposed MAS-CoPSO method. The overall objective is to clean up the polluted sea area in the most efficient way, i.e., as soon as possible. The system is supposed to realize interactive spill simulation and response optimization while considering the influences of the oceanic environment. It should also give the most reasonable allocations for available resources and the schedule of response teams' actions.

## Implementation

The implementation of MAS-CoPSO in the system is described in Fig 6, which is an instantiation of Fig 3. The MAS module and the CoPSO module work collectively, composing a dynamic system for marine oil spill response planning. The CoPSO algorithm is encapsulated as a function to be called before each time step. Meanwhile, agents of the simulation module consistently update their behaviors and interact with each other, realizing the oil spill fate modeling and response operation modeling. The agents must follow specific rules in reality. For instance, the property changes of oil slicks are calculated by weathering models considering spreading, evaporation, emulsification, and dispersion. Skimmers and ships of response teams should obey the rules for oil recovery. The performance of a response team can be affected by the actions of other teams and the characteristics of oil slicks. For example, when a skimmer works on a slick to collect oil, the volume and area of the slick, the evaporated and dispersed oil rates, the viscosity and the water content of the oil, will all be affected. If more than one team are instructed to head for the same slick, they will collaborate.

To complete the global objective of minimizing response time while cleaning up the spill site to a large extent (the stop criteria, only a small portion of leaked oil remained in the ocean), CoPSO is utilized to maximize the decreased volume of oil in each simulation time step. Meanwhile, MAS should simulate equipment location, response process, oil spill

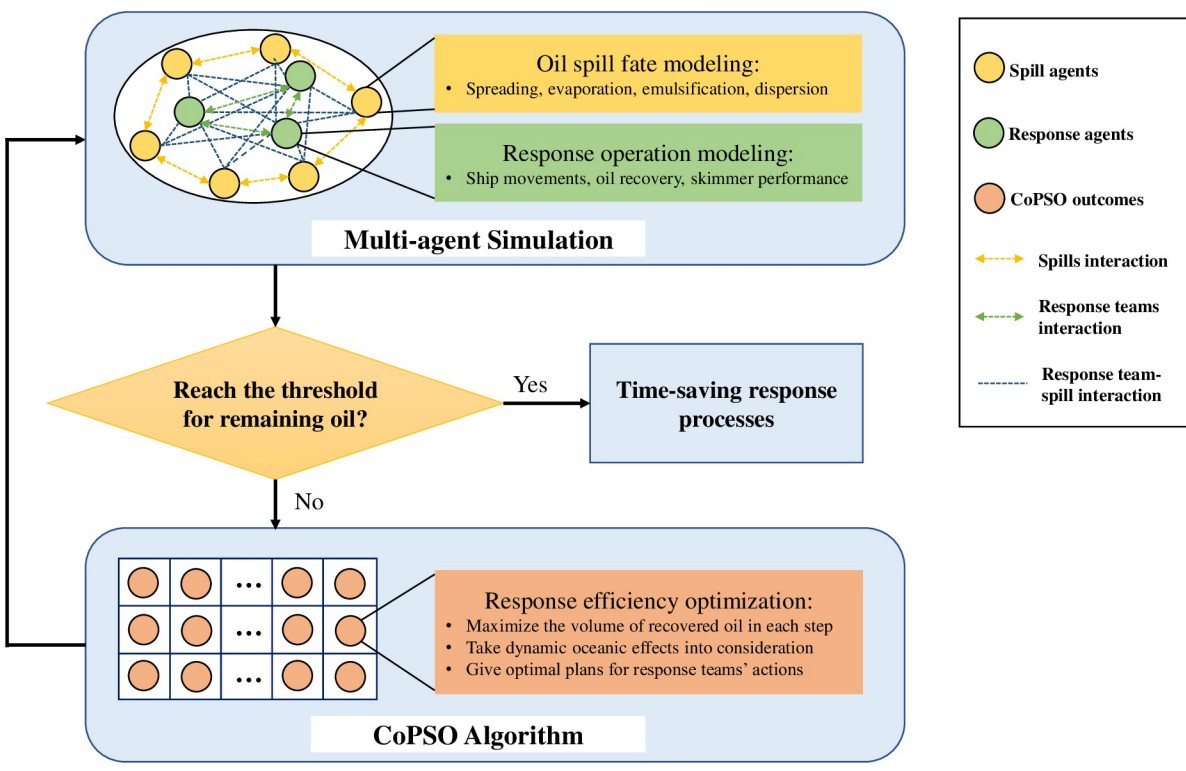

**Fig 6. Implementation of the MAS-CoPSO approach in the case study.**

fate, and transportation. The system can be further updated to suit different purposes and requirements.

## Mathematical formulation of the optimization problem

Since the weathering process dramatically affects oil properties, it is necessary to simulate it as accurately as possible. Furthermore, the objective function and constraints of the optimization problem can be derived from the model.

**Simulation of the weathering process.** First of all, the initial area of a slick, $A_0$ ($m^2$), can be calculated by the equation shown as follows [26]:

$$A_0 = \pi \frac{k_2^4}{k_3^2} \left[ \frac{(\rho_\omega - \rho_0)g V_0^5}{\rho_\omega v_\omega} \right]^{1/6}, \tag{3}$$

where $\rho_\omega$ ($g \cdot cm^{-3}$) is the density of seawater, $\rho_0$ ($g \cdot cm^{-3}$) is the density of oil, $v_\omega$ ($0.801 \times 10^{-6} m^2 \cdot s^{-1}$ under $30°C$) is the kinematic viscosity of seawater, $V_0$ ($m^3$) is the inital volume of the slick, $g$ ($m \cdot s^{-1}$) is the acceleration of gravity, $k_2$ and $k_3$ are constants with values of 1.21 and 1.53, respectively.

The area changing rate of the slick due to spreading can be modeled as follows [27]:

$$\frac{dA}{dt} = K_1 A^{-1} V^{4/3}, \tag{4}$$

where $A$ ($m^2$) is the current surface area of the slick, $t$ ($s$) refers to the time since the accident happened, $V$ ($m^3$) is the current volume of the slick, $K_1$ ($s^{-1}$) is a dominant physicochemical parameter of the oil with a default value of 150.

Evaporation is assumed as the most influential environmental effect, causing a loss of up to 20-50% of the spill. The rate that oil evaporates from the sea surface is modeled by the following equation [28]:

$$\frac{dF_E}{dt} = \frac{K_{ev}A}{V} \cdot \exp(A_{ev} - \frac{B_{ev}}{T_K} \cdot (T_O + T_G F_E)), \tag{5}$$

where $F_E$ (%) is the current volume fraction of the oil that has been evaporated with an initial value of $F_{E(t=0)} = 0$, $T_K$ (K) is the oil temperature, $A_{ev}$ and $B_{ev}$ are empirical constants with fixed values of 6.3 and 10.3, respectively. $K_{ev}$ ($m \cdot s^{-1}$) is the mass transfer coefficient for evaporation which can be calculated as follows [29]:

$$K_{ev} = 2.5 \times 10^{-3} v_{wind}^{0.78},$$

where $v_{wind}$ ($m \cdot s^{-1}$) is the wind speed. $T_O$ and $T_G$ are the initial boiling point and the gradient of the oil distillation curve, respectively. Their values can be calculated through functions of the oil API (American Petroleum Institute) degree as follows:

$$T_O = 457.16 - 3.3447 \cdot API,$$

$$T_G = 1356.7 - 247.36 \cdot \ln API.$$

Emulsification can also influence the slick characteristics dramatically, especially for the oil viscosity. The initial oil viscosity $\mu_0$ (%) can be calculated using the following equation [29]:

$$\mu_0 = 224 \times \sqrt{AC}, \tag{6}$$

where $AC$ (%) is the asphaltene content of the parent oil, which is a constant. As the viscosity changes over time, its changing rate is given by [30]:

$$\frac{d\mu}{dt} = \left[\frac{2.5\mu}{(1 - C_3 Y_W)^2}\right] \cdot \frac{dY_W}{dt} + C_4 \mu \cdot \frac{dF_E}{dt}, \tag{7}$$

where $\mu$ (%) is the current viscosity, $C_3$ is a constant for the final water content, $C_4$ is an oil-dependent constant, $Y_W$ (%) is the fractional water content in the emulsion, it changes over time, which can be computed with the following equation [27]:

$$\frac{dY_W}{dt} = K_{em} \cdot (v_{wind} + 1)^2 \cdot (1 - \frac{Y_W}{C_3}),$$

where $K_{em}$ is an empirical constant between $1 \times 10^{-6}$ and $2 \times 10^{-6}$, $Y_W$ has an initial value of $Y_{W(t=0)} = 0$.

Finally, dispersion is also a vital factor in the loss of oil. The volume of oil naturally dispersed changes over time, which can be modeled by [27]:

$$\frac{dV_D}{dt} = \frac{0.11 \cdot (v_{wind} + 1)^2 \cdot A \cdot V}{A + 50\zeta_t \cdot V \cdot \mu^{1/2}}, \tag{8}$$

where $V_D$ ($m^3$) is the total dispersed oil volume until now with an initial value of $V_{D(t=0)} = 0$ and $\zeta_t$ (cP) is the oil-water interfacial tension.

**Objective function and constraints.** In the response process, the optimization problem is formulated on the basis of each simulation time step. Suppose that the $i$th response team is recovering oil on the $k$th slick during time step $T_{step}$, define $ORR_i$ ($m^3 \cdot hr^{-1}$) as the oil recovery rate of team $i$ (i.e., the amount of oil that the team can recover per hour) and $ST_k$ ($m$) as the

current oil thickness of slick $k$. $ORR_i$ is determined by the properties of the team's skimmer and $ST_k$:

$$ORR_i = \alpha \times ST_k^2 + \beta \times ST_k, \tag{9}$$

where $\alpha$ and $\beta$ are empirical coefficients obtained from experimental tests, reflecting the features of the skimmer. $ST_k$ can be calculated by:

$$ST_k = \frac{V_k}{A_k}, \tag{10}$$

where $V_k$ ($m^3$) and $A_k$ ($m^2$) are the current oil volume and the area of the $k$th slick, respectively. $A_k$ can be calculated according to Eqs (3) and (4), $V_k$ can be computed as follows:

$$V_k = V_{k0} - (EV_k + DV_k + SV_i), \tag{11}$$

where $V_{k0}$ ($m^3$) is the remaining oil volume of this slick at the end of the last time step (i.e., the initial oil volume of the current time step), $EV_k$ ($m^3$) is the volume loss caused by evaporation, $DV_k$ ($m^3$) is the volume loss caused by dispersion, and $SV_i$ ($m^3$) is the volume recovered by team $i$. Notice that $SV_i$ may not exist, which indicates that no response team is working on slick $k$ right now. The system can get the value of $EV_k$, $DV_k$, and $SV_i$ according to Eqs (5), (8) and (9), respectively. During the simulation, for simplification, a short $T_{step}$ should be chosen and the system will use the value of $V_{k0}$ for the calculations regarding $V_k$.

The aim of the system is to clean up the polluted sea area as soon as possible. Therefore, based on the proposed method, given a fixed simulation time step, the oil volume loss will be maximized during each step, which is composed of the evaporation loss, the dispersion loss and the recovered volume. Suppose there are $m$ oil slicks and $n$ response teams during $T_{step}$, $V_{step}$ represents the total oil volume loss in this period. $F_{Ek}$ and $\mu_k$ refer to the evaporated oil fraction and the oil viscosity of slick $k$, respectively. The objective function and constraints are given by:

$$\max \quad V_{step} = \sum_{k=1}^{m} EV_k + \sum_{k=1}^{m} DV_k + \sum_{i=1}^{n} SV_i \tag{12}$$

s.t.

$$EV_k = F_{Ek} \cdot V_k, \tag{13}$$

$$\frac{dF_{Ek}}{dt} = \frac{K_{ev} A_k}{V_k} \cdot \exp(A_{ev} - \frac{B_{ev}}{T_K} \cdot (T_O + T_G F_E)), \tag{14}$$

$$F_{Ek(t=0)} = 0, \tag{15}$$

$$\frac{dDV_k}{dt} = \frac{0.11 \cdot (v_{wind} + 1)^2 \cdot A_k \cdot V_k}{A_k + 50\zeta_t \cdot V_k \cdot \mu_k^{1/2}}, \tag{16}$$

$$DV_{k(t=0)} = 0, \tag{17}$$

$$SV_i = ORR_i \times T_{step}. \tag{18}$$

## Performance of the system

To examine the efficiency of MAS-CoPSO, its performance is compared with the shortest distance selection approach (SDS). Ye et al. [31] have suggested that SDS is a simple strategy commonly used in marine oil spill emergency response, which indicates a process that allows a response team to choose the nearest oil slick as the target for oil recovery. After meeting the cleaning requirements, they will choose the second nearest oil slicks to continue.

Therefore, when conducting SDS, the distances between ships and slicks are the judgment criteria for response planning. When initializing the simulation process, each response team (corresponding to a certain response agent) is assigned with the closest oil slick (corresponding to a certain spill agent) to clean. Before each time step, instead of calling the CoPSO module, SDS will check whether the slick is cleaned up, if the oil volume is below the set threshold, the ship will head to the closest one in the remaining slicks, otherwise, it will stay at the current location until finishing the cleaning task. Compared to MAS-CoPSO, SDS is more straightforward, which explains why it is commonly used in real accidents. However, SDS makes plans without considering the oceanic forces and the interactions between agents, which according to previous discussions, can dramatically influence the overall efficiency. Thus, comparative experiments are conducted for further analysis.

**Simulation settings.**    Chen et al. [32] introduced the process and consequences of the accident in detail. Based on their description, it is assumed that the spilled oil was split into ten slicks within the East Sea with a total volume of 113,000 *tons*. Table 4 lists the oil volumes and locations (described by the distance and direction from the approximate collision site) of these slicks. The ship-mounted skimmers belonging to five different response teams are the only available nearby cleanup means to be applied. Assume that all the teams have enough storage space for recovered oil. As the allocation process needs specific transportation time, the speed of ships is set at 30 $km \cdot h^{-1}$. Skimmers of different teams have different recovering efficiencies, which are affected by $\alpha$ and $\beta$ in Eq (9). Table 5 lists the values of the empirical coefficients for different teams. Fig 7 illustrates the simulation process. The red cross represents the collision site, the black spills represent the oil slicks (the size reflects the oil volume of the corresponding slick), and the ships represent the response teams carrying the recovery task. The position and size of a slick will change over time. A ship can either be heading to a slick or working on it.

As mentioned above, spreading, evaporation, emulsification, and dispersion are the main oceanic effects. Table 6 lists the inputs for the modeling of these processes, including

**Table 4. Oil volumes and site locations of ten slicks formed after the accident.**

| Slick | Oil volume (*ton*) | Location (from the collision site) | |
|---|---|---|---|
| | | Distance (*km*) | Direction |
| 1 | 9318.66 | 14 | Northwest |
| 2 | 6593.79 | 12 | Northwest |
| 3 | 15020.19 | 11 | North |
| 4 | 8344.60 | 4 | Southwest |
| 5 | 6258.39 | 2 | North |
| 6 | 13907.59 | 2 | Southwest |
| 7 | 16689.20 | 13 | Southeast |
| 8 | 5562.99 | 2 | Southwest |
| 9 | 7253.58 | 15 | Southeast |
| 10 | 24051.01 | 0 | None |

**Table 5. Empirical coefficients for the calculations of five ship-mounted skimmers' oil recovery rates.**

| Types of skimmers | Empirical coefficients | |
|---|---|---|
| | $\alpha$ | $\beta$ |
| $SK_1(TeamA)$ | 0.01437 | 0.05602 |
| $SK_2(TeamB)$ | -0.00791 | 0.84975 |
| $SK_3(TeamC)$ | -0.01591 | 1.54975 |
| $SK_4(TeamD)$ | 0.02372 | 0.03583 |
| $SK_5(TeamE)$ | -0.01026 | 1.27589 |

environmental parameters and the characteristics of the leaked oil (condensate oil). As [26] suggested, the drifting speed of oil slicks is about 2.5-4.5% of the wind speed, so it is set at 0.03 $m \cdot s^{-1}$.

The model is built in AnyLogic®. 50 repeating runs are carried out with 50 iterations per time step ($T_{step}$ is one hour). Once the volume of the remaining oil is reduced to 10% of the original figure, the simulation will stop.

**Results and discussion.** Table 7 presents the simulation results of MAS-CoPSO and SDS with the same stop criteria and time step. The operation time for achieving an oil recovery rate

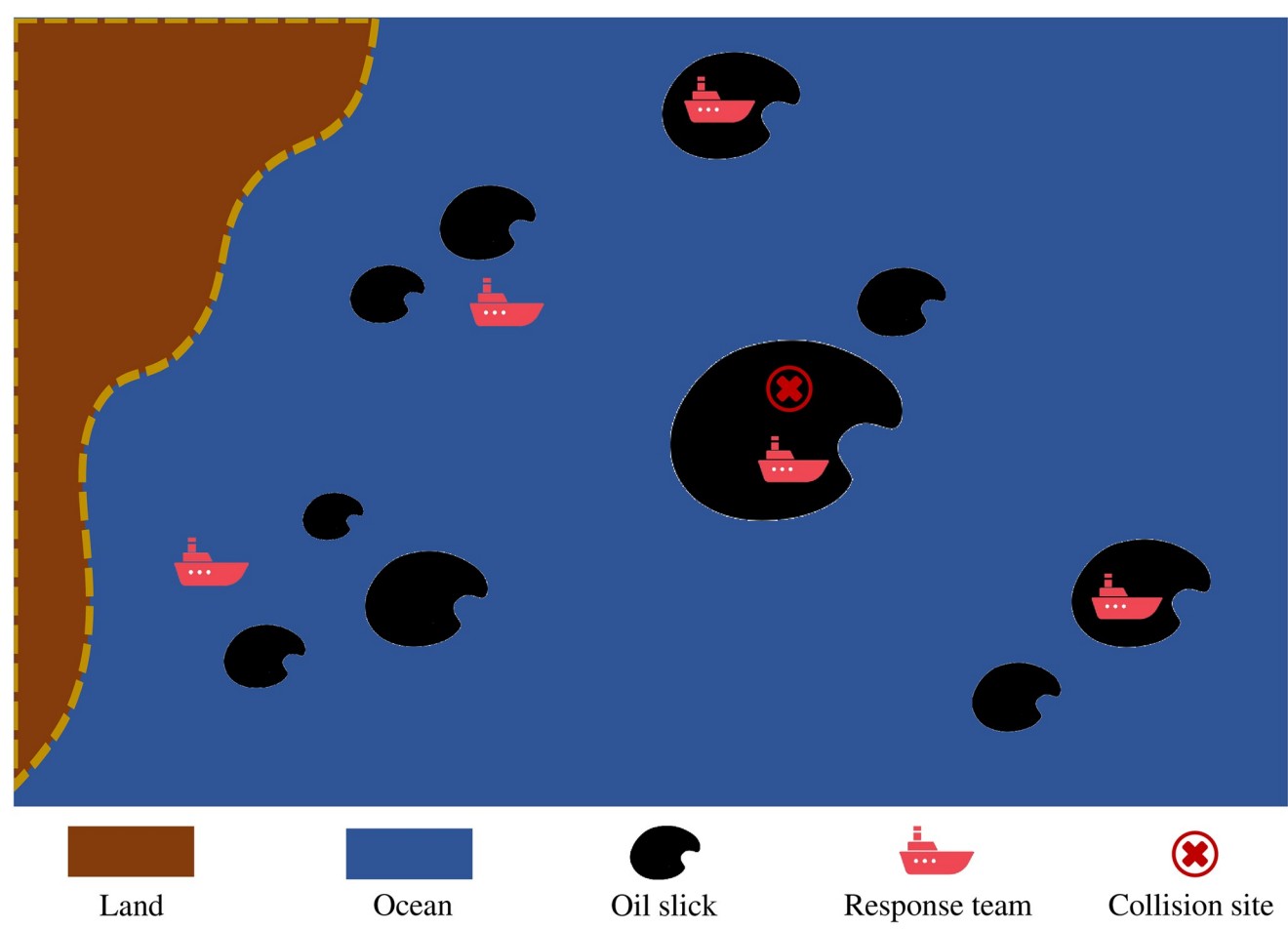

| Land | Ocean | Oil slick | Response team | Collision site |

**Fig 7. Illustration of the simulation process.**

**Table 6. Parameters for the simulation of weathering processes.**

| Parameter | Value | Unit |
|---|---|---|
| Sea water temperature ($T$) | 278.15 | $K$ |
| Wind speed ($v_{wind}$) | 10 | $m \cdot s^{-1}$ |
| Sea water density ($\rho_w$) | 1.02 | $g \cdot cm^{-3}$ |
| Interface tension ($\zeta_t$) | $2.30 \times 10^3$ | $cP$ |
| Oil density ($\rho_0$) | 0.77 | $g \cdot cm^{-3}$ |
| Oil temperature ($T_K$) | 288.15 | $K$ |
| Oil API | 52.27 | None |
| Oil AC | 1.21 | % |

of 90% is 129.82 hours based on the optimal outcomes given by the MAS-CoPSO system, which is only 74.2% of the figure for SDS. Shorter response time means higher operation efficiency, less volume of wasted oil, and slighter damage to the environment. The results also indicate that total recovered oil holds a proportion of 65.54% in MAS-CoPSO, which is approximately 1.2 times higher than the number for SDS.

Fig 8 illustrates the variations of remaining oil volume for each slick in MAS-CoPSO and SDS scenarios during the entire simulation process. The optimization outcomes of MAS-CoPSO are highly related to the thickness of the oil slicks, and the system intends to keep a balanced volume level for each slick by optimizing the time for allocation and the response efficiency (see Fig 8a). However, oil thickness does not affect the decision-making of the SDS model, so it will not try to reduce the volume steadily. As a result of different strategies, the curves of MAS-CoPSO are much smoother than those of SDS. In addition, the SDS method might ignore slicks too far from the response teams (e.g., Slick 1), leading to long-time exposure of the oil, emitting more harmful substances (see Fig 8b).

Fig 9 depicts the variations of accumulated recovered volume for each team in two approaches. MAS-CoPSO tries to find optimal schedules for all teams in every time step, so the curves climbs stably (see Fig 9a). Meanwhile, SDS always cleans up one slick before moving to another, leading to discontinuous curves, which also hinders the increase of the overall efficiency (see Fig 9b).

**Table 7. Simulation results of MAS-CoPSO and SDS.**

| | MAS-CoPSO | SDS |
|---|---|---|
| Operation time ($hr$) | **129.82** | 174.96 |
| Recovered oil (Team A) ($ton$) | 4133.35 | 2203.70 |
| Recovered oil (Team B) ($ton$) | 14658.40 | 13329.95 |
| Recovered oil (Team C) ($ton$) | 27268.71 | 20088.75 |
| Recovered oil (Team D) ($ton$) | 5598.22 | 5726.52 |
| Recovered oil (Team E) ($ton$) | 22401.52 | 20258.68 |
| Total recovered oil (%) | **65.54** | 54.52 |
| Evaporated oil (%) | 24.13 | 35.34 |
| Dispersed oil (%) | 0.52 | 0.51 |
| Remaining oil (%) | 9.81 | 9.63 |

The operation time ($hr$), the oil volume recovered by each team ($ton$), the proportion of each oil loss type (%), and the proportion of remaining oil after the response process (%) are listed in the table.

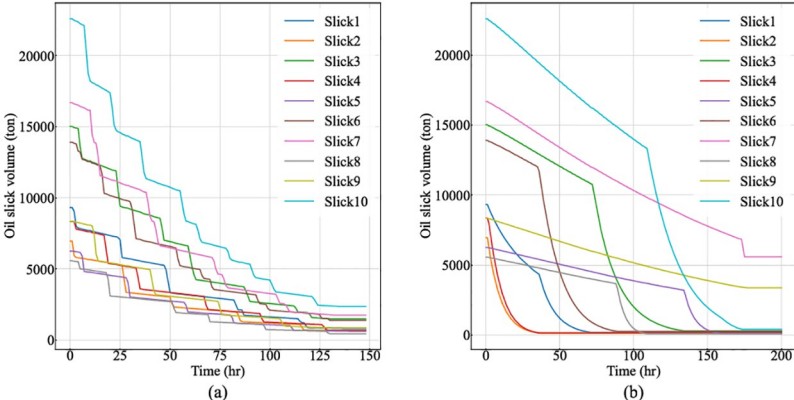

**Fig 8. Variations of remaining oil volume for each slick in MAS-CoPSO and SDS.** (a) The results of MAS-CoPSO. (b) The results of SDS.

The above results show that the response planning system based on MAS-CoPSO outperforms the commonly used SDS strategy in terms of time consumption, oil recovery rate, and environmental impact. Therefore, oil spill response teams can schedule their operations according to the simulation-optimization outcomes of this system rather than relying on the inefficient SDS approach. Moreover, complex problems and high-intensity interactions can enhance the advantages of MAS-CoPSO, indicating the system's ability to function in scenarios with higher oil volumes and more response teams. Even though the case study is simplified and focuses on the oil recovery process, the system has the potential to comprehensively support multiple cleanup techniques concerning booms, chemical dispersants, and in situ burning. Besides, modeling of the weathering process can be easily enriched by considering more complicated oceanic forces such as dissolution, photo-oxidation, sedimentation, and biodegradation [25]. Hydrodynamic simulation of oil spill trajectories can also be considered. In addition, the application range of the MAS-CoPSO-based system can be further enlarged by handling uncertainties and risk assessments. With these extensions, oil spill response teams can get a practical simulation-optimization tool to support their planning process.

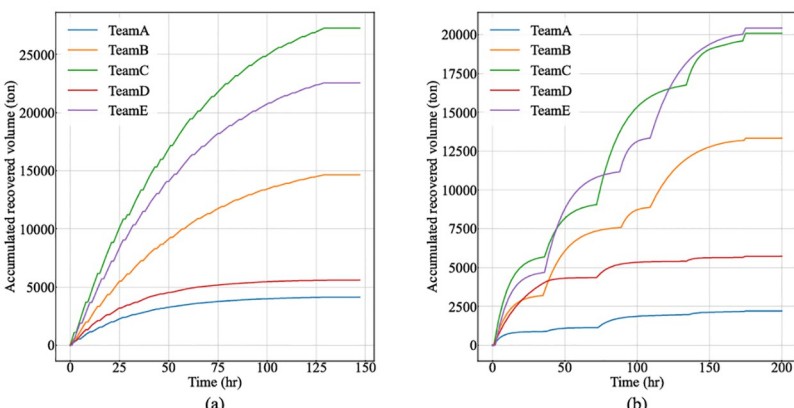

**Fig 9. Variations of accumulated recovered volume for each team in MAS-CoPSO and SDS.** (a) The results of MAS-CoPSO. (b) The results of SDS.

## Conclusion

An improved particle swarm optimization algorithm called CoPSO is proposed in this paper. Inspired by the features of bird flocks, a multi-level design and a competition mechanism are introduced into CoPSO to balance the exploration and exploitation more effectively, increase solution accuracy, achieve fast convergence, and avoid stagnation at local optima. The performance of CoPSO is tested on a series of well-known benchmark functions. Comparisons are made between CoPSO, SPSO, and three classic variants of it. Experimental results have verified its capability.

This paper also proposes a problem-independent simulation-optimization approach called MAS-CoPSO to combine CoPSO with MAS. MAS-CoPSO can achieve a dynamic simulation of various complex systems and give satisfying solutions to related optimization problems. In MAS-CoPSO, the relationship between the simulation module and the optimization module is highly cohesive. The effectiveness of MAS-CoPSO is substantiated through a response planning system for the Sanchi oil spill accident. The system simulates the accident scene with consideration of the oil slicks' physicochemical evolution. Case study results show that the system can optimize response device allocation, operation scheduling, and time consumption, causing slighter damage to the environment.

MAS-CoPSO can also be applied in other scenarios to solve different optimization problems depending on the users' requirements. For future studies, we will consider the combination with other methods, such as fuzzy control and neural network.

## Acknowledgments

We would thank Junbo Tong for his review of the manuscript.

## Author Contributions

**Conceptualization:** Shuhan Du, Wenhui Fan.

**Investigation:** Shuhan Du.

**Methodology:** Shuhan Du, Wenhui Fan.

**Software:** Shuhan Du.

**Supervision:** Wenhui Fan, Yi Liu.

**Validation:** Shuhan Du.

**Visualization:** Shuhan Du.

**Writing – original draft:** Shuhan Du.

**Writing – review & editing:** Wenhui Fan, Yi Liu.

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
