## [Decision Letter · Decision Letter 0]

20 Jun 2022

PONE-D-22-15746A novel multi-agent simulation based particle swarm optimization algorithmPLOS ONE

Dear Dr. Du,

Thank you for submitting your manuscript to PLOS ONE. After careful consideration, we feel that it has merit but does not fully meet PLOS ONE’s publication criteria as it currently stands. Therefore, we invite you to submit a revised version of the manuscript that addresses the points raised during the review process.

We look forward to receiving your revised manuscript.

Kind regards,

Ali Safaa Sadiq

Academic Editor

PLOS ONE

Journal Requirements:

2. We note that Figure 8 in your submission contain [map/satellite] images which may be copyrighted. All PLOS content is published under the Creative Commons Attribution License (CC BY 4.0), which means that the manuscript, images, and Supporting Information files will be freely available online, and any third party is permitted to access, download, copy, distribute, and use these materials in any way, even commercially, with proper attribution. For these reasons, we cannot publish previously copyrighted maps or satellite images created using proprietary data, such as Google software (Google Maps, Street View, and Earth). For more information, see our copyright guidelines: http://journals.plos.org/plosone/s/licenses-and-copyright.

a. You may seek permission from the original copyright holder of Figure 8 to publish the content specifically under the CC BY 4.0 license.  

Reviewers' comments:

Reviewer's Responses to Questions

**Comments to the Author**

1. Is the manuscript technically sound, and do the data support the conclusions?

Reviewer #1: Yes

Reviewer #2: Yes

2. Has the statistical analysis been performed appropriately and rigorously? 

Reviewer #1: Yes

Reviewer #2: Yes

3. Have the authors made all data underlying the findings in their manuscript fully available?

Reviewer #1: Yes

Reviewer #2: Yes

4. Is the manuscript presented in an intelligible fashion and written in standard English?

Reviewer #1: Yes

Reviewer #2: Yes

5. Review Comments to the Author

Reviewer #1: The contribution of this paper is good and I am happy to endorse its acceptance at some point. However, there are several major and minor comments to address. I have listed them as follows:

• First off, please clearly state the gap targeted in this paper at the end of introduction and list down the hypotheses

• In terms of research method and design, there is not much in the paper.

• The comparative algorithms in the experiments are not properly acknowledged and cited

• I also suggest adding some figures to better articular the content as the paper looks very dry at the moment.

• Analysis of the results is missing in the paper. There is a big gap between the results and conclusion. There should be the result analysis between these two sections. After comparing the numerical methods, you have to be able to analyse the results and relate them to their structures. It would be interesting to have your thoughts on why the method works that way? Such analyses would be the core of your work where you prove your understanding of the reason behind the results. You can also link the findings to the hypotheses of the paper. Long story short, this paper requires a very deep analysis from different perspectives

• There is no statistical test to judge about the significance of the numerical method’s results. Without such a statistical test, the conclusion cannot be supported

• There is no discussion on the cost effectiveness of the proposed method. What is the computational complexity? What is the runtime? Please include such discussions. You can also use the big oh notation to show the computation complexity.

• Some mathematical notations and Lemma presentations are not rigorous enough to correctly understand the contents of the paper. The authors are requested to recheck all the definition of variables and further clarify these equations.

Reviewer #2: This paper proposes a hybrid particle swarm optimisation method for multi-agent simulation case study. The paper needs some revisions and after applying them can be published.

1. please clear what are the main research gaps in Abstract.

2. The CoPSO algorithm is not clear and can be described more.

3. The major novelties of this work should be listed in the introduction.

4. Next section after introduction should be methods and materials.

5. In the section of case study, please describe the technical details of benchmarks only not methods details.

6. It can be helpful to have a better understanding of CoPSO with adding some relevant references and adaptive PSO as well such as a) FAIPSO: fuzzy adaptive informed particle swarm optimization. Neural Computing and Applications. 2013 Dec;23(1):95-116. b) "A new kind of PSO: predator particle swarm optimization." International Journal on Smart Sensing and Intelligent Systems 5, no. 2 c) An adaptive particle swarm optimizer with decoupled exploration and exploitation for large scale optimization." Swarm and Evolutionary Computation 60 (2021): 100789.

6. PLOS authors have the option to publish the peer review history of their article (what does this mean?). If published, this will include your full peer review and any attached files.

Reviewer #1: No

Reviewer #2: No

---

## [Author Response · Author response to Decision Letter 0]

16 Aug 2022

Dear Editors and Reviewers

Thank you for your careful review and constructive comments regarding our manuscript. These comments are all valuable and helpful for improving our paper. All the authors have seriously discussed about all the comments. We have tried our best to revise the manuscript in accordance with the comments and highlighted in yellow all the amends in our revised manuscript. The point-by-point replies are listed below. We sincerely appreciate your help.

Reply to Reviewer 1 Comments

Comment 1

First off, please clearly state the gap targeted in this paper at the end of introduction and list down the hypotheses.

-Response & Revision:

Thank you very much for the valuable feedback on our Introduction. We have modified the end of our Introduction as follows:

We can see from the literature that although there are many studies on improving SPSO, there are not so many attempts to combine proposed algorithms with MAS and apply them to practical problems. In fact, utilizing improved algorithms in simulation-optimization systems is necessary because complex tasks require high-performance optimizers, and SPSO may not meet this requirement. Besides, existing studies on the combination of MAS and PSO may vary in engineering fields, yet they all designed the system based on specific problems of their own research field without summarizing a general and universal framework, which is field-independent and can be implemented in various settings. Therefore, this paper focuses on a more in-depth exploration of improving PSO and integrating it into MAS to solve practical problems. The main contributions of this paper are as follows:

1. We proposed an improved particle swarm optimization algorithm called CoPSO according to the biological nature of birds. CoPSO has a multi-level structure and a competition mechanism, aiming to improve performance by enhancing exploration while balancing exploitation.

2. We proposed a general approach to integrate CoPSO into MAS called MAS-CoPSO. MAS-CoPSO is supposed to achieve a very close combination between simulation and optimization, and the application should not be limited to certain engineering fields and specific problems.

3. We conducted a series of comparison experiments to substantiate the validity of CoPSO.

4. We implemented MAS-CoPSO in a case study of the Sanchi oil spill accident to solve the response planning problem.

Comment 2

In terms of research method and design, there is not much in the paper.

-Response & Revision:

We appreciate a lot for your comments on the formation of the paper. To address this problem, we have adjusted the structure of the paper by adding section Methods and materials after Introduction. There are three subsections in Methods and materials, namely, Multi-agent Simulation, The proposed CoPSO algorithm, and The MAS-CoPSO approach. These three parts have described the research method and design of the paper in detail.

Comment 3

The comparative algorithms in the experiments are not properly acknowledged and cited. 

-Response & Revision:

Thank you for pointing out this problem. We have four comparative algorithms, namely, the standard formation of PSO (SPSO) [1], the random weight method (RPSO) [2], the constriction factor approach (CPSO) [3], and the incremental particle swarm optimizer (IPSO) [4]. We briefly mentioned these algorithms in Introduction without proper acknowledgments and citations in the experiments. For modification, we have cited these algorithms and described their contents in section Experiments and analysis of CoPSO.

Comment 4

I also suggest adding some figures to better articular the content as the paper looks very dry at the moment.

-Response & Revision:

Thank you for your constructive suggestion. Firstly, we have added Fig 1 and Fig 2 (the mark numbers subject to the revised manuscript) for a better description of CoPSO. They are both two-dimensional schematic diagrams. Fig 1 illustrates how the particles’ levels change during the iterations. Fig 2 depicts how the competition mechanism functions in the algorithm. We have also added Fig 5 in Case study of MAS-CoPSO to enrich the background description. Fig 5 is a geographical map illustrating the approximate location where the Sanchi oil spill accident happened. In addition, we have modified some original figures. Fig 4 in the revised manuscript contains subgraphs of benchmark functions’ converging behaviors and their swarm diversity variations. We have also redrawn Fig 7 to depict the simulation process of the case study more clearly. Fig 8 and Fig 9 are modified for better comparison.

Comment 5

Analysis of the results is missing in the paper. There is a big gap between the results and conclusion. There should be the result analysis between these two sections. After comparing the numerical methods, you have to be able to analyze the results and relate them to their structures. It would be interesting to have your thoughts on why the method works that way? Such analyses would be the core of your work where you prove your understanding of the reason behind the results. You can also link the findings to the hypotheses of the paper. Long story short, this paper requires a very deep analysis from different perspectives.

-Response & Revision:

Thank you for pointing out this problem. As we have adjusted the formation of the paper, modifications for this comment can be found in section Experiments and analysis of CoPSO. Considering some exceptional results may hugely affect the “Mean” performance and conceal the algorithms’ properties, we have added the comparison item “Median” in Table 2. Further, we have also introduced the concept of swarm diversity [5] to analyze the algorithms’ exploration abilities. After comparing the diversity evolution and the convergence behavior of each algorithm on all the benchmarks in Fig 4, we have discussed the results based on the architectures of the algorithms and concluded that the multi-level structure and the competition mechanism did improve the performance of CoPSO by enhancing exploration while balancing exploitation. A detailed analysis can be found in Results and discussion of Comparison experiments.

Comment 6

There is no statistical test to judge about the significance of the numerical method’s results. Without such a statistical test, the conclusion cannot be supported.

-Response & Revision:

Thank you for this feedback. For modification, as recommended by Yang et al. [5], we have conducted Wilcoxon rank sum test for the statistical analysis between CoPSO and the other algorithms with a significance level of 0.05. The p-values were listed in Table 2. Results of the statistical test have supported our conclusion. A detailed discussion can be found in Results and discussion of Comparison experiments.

Comment 7

There is no discussion on the cost effectiveness of the proposed method. What is the computational complexity? What is the runtime? Please include such discussions. You can also use the big oh notation to show the computation complexity.

-Response & Revision:

Thank you for this comment. We have added subsection Computational complexity analysis in Experiments and analysis of CoPSO to make up for this part. According to Yang et al. [5], given a fixed number of fitness evaluations, the computational complexity of an evolutionary algorithm is generally calculated by analyzing the extra cost in each generation without considering the cost of function evaluations, which is problem-dependent. As CoPSO inherits the simple structure of SPSO, we have analyzed the computational complexity (in terms of time and space, using the big oh notation) of CoPSO by comparing it with SPSO. We have also compared the runtime of all the algorithms and analyzed the results listed in Table 3.

Comment 8

Some mathematical notations and Lemma presentations are not rigorous enough to correctly understand the contents of the paper. The authors are requested to recheck all the definition of variables and further clarify these equations.

-Response & Revision:

Thank you for pointing out this problem. We have rechecked the paper and modified the following ambiguous mathematical notations and lemma presentations (the mark numbers of the tables and equations subject to the revised manuscript):

 As the position and velocity of a particle (for example, particle i) are supposed to be vectors, we have modified the presentation of these two variables to the bold form x_i=(x_i^1,x_i^2,…,x_i^D) and v_i=(v_i^1,v_i^2,…,v_i^D), where the superscripts indicate dimensions. We have also modified the personal best position of particle i and the global best position to 〖pbest〗_i=(〖pbest〗_i^1,〖pbest〗_i^2,…,〖pbest〗_i^D) and gbest=(〖gbest〗^1,〖gbest〗^2,…,〖gbest〗^D), respectively. For a more concise expression, we have modified the denotation of the level of particle i to l_i with an upper bound of L_max. Finally, we have further clarified Eq (1), Eq (2), and Algorithm 1.

 We have added the definitions of n (the problem dimension) and f_min (the global optimum) in the note of Table 1.

 We have added the units of ρ_ω and ρ_0 in Eq (3).

 We have supplemented the definitions of A, t and V in Eq (4).

 We have supplemented the definitions of F_E and added the calculation of T_O and T_G in Eq (5).

 We have added the unit of μ_0 and the explanation of AC in Eq (6), as well as the unit of μ and the calculation of Y_W in Eq (7).

 We have supplemented the definitions of V_D and added the unit of ζ_t in Eq (8). We have also modified the presentation of Eq (8) to avoid the misunderstanding of variable A and variable V.

 We have supplemented the definitions of 〖ORR〗_i in Eq (9) and 〖ST〗_k in Eq (10). We have also added the explanation of parameter α and β in Eq (9), as well as the calculation of A_k in Eq (10).

 We have supplemented the definitions and the calculations of V_k, V_k0, 〖EV〗_k, DV_k, and 〖SV〗_i in Eq (11).

 We have added the definitions of V_step in Eq (12), 〖F_E〗_k in Eq (14), and μ_k in Eq (16). We have also modified Eq (13), Eq (14), and Eq (16) for better presentation. 

 We have stated our objective in text form for a better description of Eqs (12)~(18) (The aim of our system is to clean up the polluted sea area as soon as possible. Therefore, based on our method, given a fixed simulation time step, we will maximize the oil volume loss during each step, which is composed of the evaporation loss, the dispersion loss and the recovered volume.).

 We have added the meaning of each result item in the note of Table 7.

References in replies:

[1] Kennedy J, Eberhart R. Particle swarm optimization. In: Proceedings of ICNN’95-international conference on neural networks. vol. 4. IEEE; 1995. p. 1942–1948.

[2] Eberhart RC, Shi Y. Tracking and optimizing dynamic systems with particle swarms. In: Proceedings of the 2001 congress on evolutionary computation (IEEE Cat. No. 01TH8546). vol. 1. IEEE; 2001. p. 94–100.

[3] Clerc M, Kennedy J. The particle swarm-explosion, stability, and convergence in a multidimensional complex space. IEEE transactions on Evolutionary Computation. 2002;6(1):58–73.

[4] De Oca MAM, Stutzle T, Van den Enden K, Dorigo M. Incremental social learning in particle swarms. IEEE Transactions on Systems, Man, and Cybernetics, Part B (Cybernetics). 2010;41(2):368–384.

[5] Yang Q, Chen WN, Da Deng J, Li Y, Gu T, Zhang J. A level-based learning swarm optimizer for large-scale optimization. IEEE Transactions on Evolutionary Computation. 2017;22(4):578–594.

Reply to Reviewer 2 Comments

Comment 1

Please clear what are the main research gaps in Abstract.

-Response & Revision:

Thank you very much for pointing out the problems in our Abstract. For modification, we have stated the main research gaps and the contributions of our paper as follows:

Recently, there have been considerable researches on combining multi-agent simulation and particle swarm optimization in practice. However, most existing studies are limited to specific engineering fields or problems without summarizing a general and universal combination framework. Moreover, particle swarm optimization can be less effective in complex problems due to its weakness in balancing exploration and exploitation. Yet, it is not common to integrate MAS with improved versions of the algorithm. Therefore, this paper proposed an improved particle swarm optimization algorithm called CoPSO, introducing a multi-level structure and a competition mechanism to enhance exploration while balancing exploitation. Further, we integrated CoPSO into MAS by a problem-independent approach called MAS-CoPSO, aiming to simulate realistic scenarios dynamically and solve related optimization problems simultaneously.

Comment 2

The CoPSO algorithm is not clear and can be described more.

-Response & Revision:

Thank you for this feedback. We have modified subsection Improved particle swarm optimization algorithm with competition mechanism to address this problem, a detailed description can be found there. Firstly, we have introduced the standard formation of PSO and stated our motivation, i.e., improving the performance of PSO by enhancing exploration while balancing exploitation. Inspired by the biological nature of birds, we have proposed a multi-level structure and a competition mechanism in CoPSO. We divide particles into different levels and stipulate that the closer a particle is to a possible global optimum, the higher its level. If a particle doesn't update its personal best position in μ continuous iterations, it will upgrade. Particles of different levels learn at different acceleration constants. Fig 1 is a two-dimensional schematic diagram of the variations of particles’ levels during the iterations.

However, a higher level doesn’t always a mean closer distance to the global optimum, it can also indicate that the particle has fallen into local optima. The basic idea of the competition mechanism is to replace particles of extremely high levels with randomly generated new ones. Thus, the algorithm gets the potential to escape from local optima and explore the unknown search space. Fig 2 illustrates how the competition mechanism functions, particles in Fig 2a are trapped in local areas, once their levels get too high, we will regenerate new particles to replace them. From Fig 2b we can see that these new particles have located the right promising area. 

The pseudo-code of CoPSO is shown in Algorithm 1.

Comment 3

The major novelties of this work should be listed in the introduction.

-Response & Revision:

Thank you for pointing out this problem. We have listed the novelties of the paper in the end of our Introduction as follows:

The main contributions of this paper are as follows:

1. We proposed an improved particle swarm optimization algorithm called CoPSO according to the biological nature of birds. CoPSO has a multi-level structure and a competition mechanism, aiming to improve performance by enhancing exploration while balancing exploitation.

2. We proposed a general approach to integrate CoPSO into MAS called MAS-CoPSO. MAS-CoPSO is supposed to achieve a very close combination between simulation and optimization, and the application should not be limited to certain engineering fields and specific problems.

3. We conducted a series of comparison experiments to substantiate the validity of CoPSO.

4. We implemented MAS-CoPSO in a case study of the Sanchi oil spill accident to solve the response planning problem.

Comment 4

Next section after introduction should be methods and materials.

-Response & Revision:

Thank you for your constructive comments on the formation of the paper. We have added section Methods and materials after Introduction. There are three subsections in Methods and materials, namely, Multi-agent Simulation, The proposed CoPSO algorithm, and The MAS-CoPSO approach. These three parts have described the research method and design of the paper in detail.

Comment 5

In the section of case study, please describe the technical details of benchmarks only not methods details.

-Response & Revision:

Thank you for the valuable feedback. Based on the modeling of an oil spill accident, we conducted MAS-CoPSO for response planning and used the shortest distance selection approach (SDS) as a contrast. SDS is commonly used in real oil spill response emergencies [1], which makes the comparison reasonable. We have added a more detailed description of the procedure of SDS in subsection The performance of the MAS-CoPSO-based system. Instead of calling the CoPSO module, SDS used the distances between ships and oil slicks as the measure for planning, which was the main difference between these two approaches. When initializing the simulation process, we assigned each response team with the closest oil slick to clean. Before each time step, we would check whether the slick was cleaned up, if the oil volume was below the set threshold, the ship would head to the closest one in the remaining slicks, otherwise, it would stay at the current location until completing the recovery.

Comment 6

It can be helpful to have a better understanding of CoPSO with adding some relevant references and adaptive PSO as well such as a) FAIPSO: fuzzy adaptive informed particle swarm optimization. Neural Computing and Applications. 2013 Dec;23(1):95-116. b) "A new kind of PSO: predator particle swarm optimization." International Journal on Smart Sensing and Intelligent Systems 5, no. 2 c) An adaptive particle swarm optimizer with decoupled exploration and exploitation for large scale optimization." Swarm and Evolutionary Computation 60 (2021): 100789.

-Response & Revision:

Thank you for providing these papers. We have cited it in our Introduction. 

References in replies:

[1] Ye X, Chen B, Li P, Jing L, Zeng G. A simulation-based multi-agent particle swarm optimization approach for supporting dynamic decision making in marine oil spill responses. Ocean & Coastal Management. 2019;172:128–136.

[2] Neshat M. FAIPSO: fuzzy adaptive informed particle swarm optimization. Neural Computing and Applications. 2013;23(1):95–116.

[3] Neshat M, Sargolzaei M, Masoumi A, Najaran A. A new kind of PSO: predator particle swarm optimization. International Journal on Smart Sensing and Intelligent Systems. 2012;5(2).

[4] Li D, Guo W, Lerch A, Li Y, Wang L, Wu Q. An adaptive particle swarm optimizer with decoupled exploration and exploitation for large scale optimization. Swarm and Evolutionary Computation. 2021;60:100789.

---

## [Decision Letter · Decision Letter 1]

29 Aug 2022

PONE-D-22-15746R1A novel multi-agent simulation based particle swarm optimization algorithmPLOS ONE

Dear Dr. Du,

Thank you for submitting your manuscript to PLOS ONE. After careful consideration, we feel that it has merit but does not fully meet PLOS ONE’s publication criteria as it currently stands. Therefore, we invite you to submit a revised version of the manuscript that addresses the points raised during the review process.

We look forward to receiving your revised manuscript.

Kind regards,

Ali Safaa Sadiq

Academic Editor

PLOS ONE

Journal Requirements:

Additional Editor Comments:

Authors are invited to address some minor comments given by the first reviewer and produce a response letter detailing out the changes made accordingly.

Reviewers' comments:

Reviewer's Responses to Questions

**Comments to the Author**

1. If the authors have adequately addressed your comments raised in a previous round of review and you feel that this manuscript is now acceptable for publication, you may indicate that here to bypass the “Comments to the Author” section, enter your conflict of interest statement in the “Confidential to Editor” section, and submit your "Accept" recommendation.

Reviewer #1: (No Response)

Reviewer #2: All comments have been addressed

2. Is the manuscript technically sound, and do the data support the conclusions?

Reviewer #1: (No Response)

Reviewer #2: Yes

3. Has the statistical analysis been performed appropriately and rigorously? 

Reviewer #1: (No Response)

Reviewer #2: Yes

4. Have the authors made all data underlying the findings in their manuscript fully available?

Reviewer #1: (No Response)

Reviewer #2: Yes

5. Is the manuscript presented in an intelligible fashion and written in standard English?

Reviewer #1: (No Response)

Reviewer #2: Yes

6. Review Comments to the Author

Reviewer #1: Some final cosmetic comments:

* The results of your comparative study should be discussed in-depth and with more insightful comments on the behaviour of your algorithm on various case studies. Discussing results should not mean reading out the tables and figures once again.

* Avoid lumping references as in [x, y] and all other. Instead summarize the main contribution of each referenced paper in a separate sentence. For scientific and research papers, it is not necessary to give several references that say exactly the same. Anyway, that would be strange, since then what is innovative scientific contribution of referenced papers? For each thesis state only one reference.

* Avoid using first person.

* Avoid using abbreviations and acronyms in title, abstract, headings and highlights.

* Please avoid having heading after heading with nothing in between, either merge your headings or provide a small paragraph in between.

* The first time you use an acronym in the text, please write the full name and the acronym in parenthesis. Do not use acronyms in the title, abstract, chapter headings and highlights.

* The results should be further elaborated to show how they could be used for the real applications.

Reviewer #2: The authors have sufficiently addressed the reviewed issues in the manuscript and it can be published now.

7. PLOS authors have the option to publish the peer review history of their article (what does this mean?). If published, this will include your full peer review and any attached files.

Reviewer #1: No

Reviewer #2: No

---

## [Author Response · Author response to Decision Letter 1]

8 Sep 2022

Dear Reviewers:

Thank you for your careful review and constructive comments regarding our manuscript. These comments are all valuable and helpful for improving our paper. All the authors have seriously discussed about all the comments. We have tried our best to revise the manuscript in accordance with the comments and highlighted in yellow all the amends in our revised manuscript. The point-by-point replies are listed below. We sincerely appreciate your help.

Reply to Reviewer 1 Comments

Comment 1

The results of your comparative study should be discussed in-depth and with more insightful comments on the behavior of your algorithm on various case studies. Discussing results should not mean reading out the tables and figures once again.

-Response & Revision:

Thank you very much for the valuable feedback. We have modified the discussion of the comparative study results in the revised manuscript as follows:

Generally, CoPSO converges faster with better solutions than all other algorithms in most cases. As for swarm diversity, CoPSO holds a similar tendency on all benchmarks, i.e., decreasing rapidly in the early stage of the evolution and staying at a relatively high value (or even the highest) in the latter stage. This can be explained by the structure of CoPSO. Particles of different levels are in different regions of the search space. Those of lower levels are more likely to be far away from the global best position while those of higher levels can be very close to it. CoPSO treats different levels differently by enhancing exploration for lower levels and promoting exploitation for higher ones. Thus, the algorithm has more potential to locate promising areas in the early stage and then converges to them very quickly, causing the swarm diversity to decline. However, particles may not upgrade because they find a global optimum, but because they are trapped in local optima. In this case, the competition mechanism will force the swarm to escape from local areas, which increases the diversity simultaneously.

Despite the overall similarity, the behaviors of CoPSO differ slightly on different kinds of test functions. f_1 is a simple unimodal function with only one global minimum, making it possible to achieve fast convergence. Although all algorithms converge exponentially to the optima, CoPSO greatly surpasses the others due to better exploitation of the promising areas at the beginning and high-intensity exploration in the latter stage. For f_2, CoPSO has a competitive performance compared to RPSO and even shows better potential in the end as a result of higher swarm diversity. f_3 is a noisy quartic function. f_4 is a classic optimization problem with a global minimum inside a long, narrow, parabolic-shaped valley. f_5 is a multimodal function where the number of local optima increases exponentially as the problem dimension increases. For all these complicated benchmarks, CoPSO always keeps an excellent balance between exploration and exploitation, which helps it to achieve the fastest converging rate or the shortest stagnation at local optima. The other algorithms lack the adjusting ability of CoPSO. For instance, IPSO puts too much emphasis on exploration (holding the highest swarm diversity on most benchmarks) while RPSO and CPSO are too focused on exploitation, which degenerates their overall performances.

Comment 2

Avoid lumping references as in [x, y] and all other. Instead summarize the main contribution of each referenced paper in a separate sentence. For scientific and research papers, it is not necessary to give several references that say exactly the same. Anyway, that would be strange, since then what is innovative scientific contribution of referenced papers? For each thesis state only one reference.

-Response & Revision:

Thank you for pointing out this problem. We have checked the references and removed items containing the same information as those kept in the revised manuscript (i.e., references 21, 23, 25, 26, and 27 of the original version).

Comment 3

Avoid using first person.

-Response & Revision:

Thank you for pointing out this problem. We have carefully checked the whole manuscript and modified related content.

Comment 4

Avoid using abbreviations and acronyms in title, abstract, headings and highlights.

-Response & Revision:

Thank you for your constructive suggestion. We have carefully checked these parts and modified related expressions to avoid using abbreviations and acronyms.

Comment 5

Please avoid having heading after heading with nothing in between, either merge your headings or provide a small paragraph in between.

-Response & Revision:

Thank you for pointing out this problem. We have noticed that this problem exists at the beginning of some sections and subsections. For modification, we have provided brief summaries in between. Please review the rebuttal letter or the revised manuscript for details.

Comment 6

The first time you use an acronym in the text, please write the full name and the acronym in parenthesis. Do not use acronyms in the title, abstract, chapter headings and highlights.

-Response & Revision:

Thank you for this feedback. We have carefully checked the manuscript and modified related content.

Comment 7

The results should be further elaborated to show how they could be used for the real applications.-Response & Revision:

Thank you for this comment. Our method has the potential to be used for real applications mainly because: 1) case study results indicate its superiority over SDS (which is commonly used in real situations); 2) it can still function in more complex scenarios; 3) it can be easily extended according to the users’ requirements and achieve a comprehensive simulation of reality. We have further discussed the results in the revised manuscript as follows:

The above results show that the response planning system based on MAS-CoPSO outperforms the commonly used SDS strategy in terms of time consumption, oil recovery rate, and environmental impact. Therefore, oil spill response teams can schedule their operations according to the simulation-optimization outcomes of this system rather than relying on the inefficient SDS approach. Moreover, complex problems and high-intensity interactions can enhance the advantages of MAS-CoPSO, indicating the system's ability to function in scenarios with higher oil volumes and more response teams. Even though the case study is simplified and focuses on the oil recovery process, the system has the potential to comprehensively support multiple cleanup techniques concerning booms, chemical dispersants, and in situ burning. Besides, modeling of the weathering process can be easily enriched by considering more complicated oceanic forces such as dissolution, photo-oxidation, sedimentation, and biodegradation [25]. Hydrodynamic simulation of oil spill trajectories can also be considered. In addition, the application range of the MAS-CoPSO-based system can be further enlarged by handling uncertainties and risk assessments. With these extensions, oil spill response teams can get a practical simulation-optimization tool to support their planning process.

---

## [Decision Letter · Decision Letter 2]

12 Sep 2022

PONE-D-22-15746R2A novel multi-agent simulation based particle swarm optimization algorithmPLOS ONE

Dear Dr. Du,

Thank you for submitting your manuscript to PLOS ONE. After careful consideration, we feel that it has merit but does not fully meet PLOS ONE’s publication criteria as it currently stands. Therefore, we invite you to submit a revised version of the manuscript that addresses the points raised during the review process.

We look forward to receiving your revised manuscript.

Kind regards,

Ali Safaa Sadiq

Academic Editor

PLOS ONE

Journal Requirements:

Additional Editor Comments:

Authors are invited to response to some of the minor comments given by the first reviewer.

Reviewers' comments:

Reviewer's Responses to Questions

**Comments to the Author**

1. If the authors have adequately addressed your comments raised in a previous round of review and you feel that this manuscript is now acceptable for publication, you may indicate that here to bypass the “Comments to the Author” section, enter your conflict of interest statement in the “Confidential to Editor” section, and submit your "Accept" recommendation.

Reviewer #1: (No Response)

2. Is the manuscript technically sound, and do the data support the conclusions?

Reviewer #1: (No Response)

3. Has the statistical analysis been performed appropriately and rigorously? 

Reviewer #1: (No Response)

4. Have the authors made all data underlying the findings in their manuscript fully available?

Reviewer #1: (No Response)

5. Is the manuscript presented in an intelligible fashion and written in standard English?

Reviewer #1: (No Response)

6. Review Comments to the Author

Reviewer #1: Excellent effort on the revision. Just one final check:

• Are all the images used in this work copyrights free? If not, have the authors obtained proper copyrights permission to re-use them? Please kindly clarify, and this is just to ensure all the figures are fine to be published in this work.

• Also, the list of references should be carefully checked to ensure consistency with between all references and their compliances with the journal policy on referencing.

7. PLOS authors have the option to publish the peer review history of their article (what does this mean?). If published, this will include your full peer review and any attached files.

Reviewer #1: No

---

## [Author Response · Author response to Decision Letter 2]

12 Sep 2022

Dear Editors and Reviewers:

Thank you for your careful review and constructive comments regarding our manuscript. These comments are all valuable and helpful for improving our paper. All the authors have seriously discussed about all the comments. We have tried our best to revise the manuscript in accordance with the comments. The point-by-point replies are listed below. We sincerely appreciate your help.

Reply to Reviewer 1 Comments

Comment 1

Are all the images used in this work copyrights free? If not, have the authors obtained proper copyrights permission to re-use them? Please kindly clarify, and this is just to ensure all the figures are fine to be published in this work.

-Response & Revision:

Thank you very much for the comment. We have carefully checked the images used in this work. All the images are copyrights free.

Comment 2

Also, the list of references should be carefully checked to ensure consistency with between all references and their compliances with the journal policy on referencing.

-Response & Revision:

Thank you for the suggestion. We have checked the reference list according to the manuscript content and the journal policy. The reference list is complete and correct. None of the cited papers have been retracted.

---

## [Decision Letter · Decision Letter 3]

26 Sep 2022

A novel multi-agent simulation based particle swarm optimization algorithm

PONE-D-22-15746R3

Dear Dr. Du,

We’re pleased to inform you that your manuscript has been judged scientifically suitable for publication and will be formally accepted for publication once it meets all outstanding technical requirements.

Kind regards,

Ali Safaa Sadiq

Academic Editor

PLOS ONE

Additional Editor Comments (optional):

I am happy to accept the manuscript for the possible publication as the authors have addressed all the given comments by the reviewers.

Reviewers' comments:

Reviewer's Responses to Questions

**Comments to the Author**

1. If the authors have adequately addressed your comments raised in a previous round of review and you feel that this manuscript is now acceptable for publication, you may indicate that here to bypass the “Comments to the Author” section, enter your conflict of interest statement in the “Confidential to Editor” section, and submit your "Accept" recommendation.

Reviewer #1: (No Response)

2. Is the manuscript technically sound, and do the data support the conclusions?

Reviewer #1: (No Response)

3. Has the statistical analysis been performed appropriately and rigorously? 

Reviewer #1: (No Response)

4. Have the authors made all data underlying the findings in their manuscript fully available?

Reviewer #1: (No Response)

5. Is the manuscript presented in an intelligible fashion and written in standard English?

Reviewer #1: (No Response)

6. Review Comments to the Author

Reviewer #1: good improvement.good improvement.good improvement.good improvement.good improvement.good improvement.good improvement.good improvement.

7. PLOS authors have the option to publish the peer review history of their article (what does this mean?). If published, this will include your full peer review and any attached files.

Reviewer #1: No

---

## [Editor Report · Acceptance letter]

3 Oct 2022

PONE-D-22-15746R3 

A novel multi-agent simulation based particle swarm optimization algorithm 

Dear Dr. Du:

I'm pleased to inform you that your manuscript has been deemed suitable for publication in PLOS ONE. Congratulations! Your manuscript is now with our production department. 

Kind regards, 

on behalf of

Dr. Ali Safaa Sadiq 

Academic Editor

PLOS ONE